# Magnetic particle imaging angiography of the femoral artery in a human cadaveric perfusion model
Viktor Hartung [1] ✉, Philipp Gruschwitz [1], Anne Marie Augustin[1], Jan-Peter Grunz [1,2], Florian Kleefeldt[3], Dominik Peter[4], Süleyman Ergün [3], Johanna Günther [5], Teresa Reichl[5], Thomas Kampf[5,6], Martin Andreas Rückert[5], Stefan Herz[1,7], Volker Christian Behr[5], Thorsten Alexander Bley[1] & Patrick Vogel [5,8]

## Abstract

**Background** Magnetic particle imaging (MPI) allows for radiation-free visualization of tracers without background signal. With the first human-sized interventional MPI scanner being recently developed, the aim of the present study was to test its performance for guiding of endovascular procedures in a realistic perfusion model. **Methods** Three fresh-frozen cadaveric legs were prepared to establish continuous circulation in the superficial femoral artery via introducer sheaths in the inguinal and infragenicular region. To facilitate vessel visualization, a mixture of a MPI tracer (Resotran® or Perimag®) and X-ray contrast agent was injected under continuous extracorporeal perfusion and imaged simultaneously with MPI angiography and digital subtraction angiography (DSA) as reference. **Results** The MPI scanner integrates seamlessly into the standard operating procedures in the angiography suite and simultaneous imaging with DSA and MPI is feasible. The MPI scanner detects a tracer bolus of 2 ml Perimag® or 1.5 ml Resotran®. Imaging results are consistent and reproducible in three cadaveric leg phantoms. **Conclusion** This study demonstrates, that the recently developed human-sized MPI scanner facilitates reliable radiation-free image guidance for peripheral vascular interventions in the superficial femoral artery with a tracer approved for use in humans.

## Plain language summary

Doctors often use imaging techniques to enable procedures to be undertaken inside blood vessels without having to use major surgery. However, many of these methods involve radiation. Magnetic Particle Imaging (MPI) can provide clear images without using radiation. We used a MPI scanner on legs from deceased humans and showed we could visualize the femoral artery following injection with a blood vessel tracer. Our results show that MPI could be used to guide procedures instead of traditional X-ray imaging. Thus MPI could become a safer alternative to guide blood vessel interventions in the future.

Peripheral arterial disease is caused by symptomatic narrowing of vessels due to atherosclerotic deposits. There is a large overlap with cardiovascular and cerebrovascular diseases such as myocardial infarction and stroke. Despite its frequent occurrence, the disease is often underdiagnosed and therefore inadequately treated, with the associated burden on the healthcare system[1].

If the diagnosis has been established by non-invasive methods, such as Doppler examinations (Ankle-brachial index) or imaging and if conservative treatment methods have been exhausted, percutaneous transluminal angioplasty is the standard therapy for lower extremity peripheral

arterial disease[2]. The number of vascular interventions in Germany has doubled in the last decade to approximately 300,000 in 2021[3]. To date, ionizing radiation has always been necessary for image guidance of endovascular procedures. Low levels of ionizing radiation carry a substantial risk in excess cancer mortality and cardiovascular disease among healthcare workers and patients alike[4–6]. The German national reference guidelines allow an effective dose of 10 mSv for examinations of the thigh[7], which corresponds to about 5 years of natural background radiation. Although radiation dose efficiency increases with new X-ray detector generations and improved post-processing algorithms, these effects are only incremental[8].

[1]Department of Diagnostic and Interventional Radiology, University Hospital Würzburg, Würzburg, Germany. [2]Department of Radiology, University of Wisconsin, Madison, WI, USA. [3]Institute of Anatomy and Cell Biology, Julius-Maximilians University, Würzburg, Germany. [4]Department of General, Visceral, Transplant, Vascular and Pediatric Surgery, Center of Operative Medicine, University Hospital Würzburg, Würzburg, Germany. [5]Department of Experimental Physics 5 (Biophysics), Julius-Maximilians University, Würzburg, Germany. [6]Department of Diagnostic and Interventional Neuroradiology, University Hospital Würzburg, Würzburg, Germany. [7]Radiologie Augsburg Friedberg, Augsburg, Germany. [8]Pure Devices GmbH, Rimpar, Germany. ✉e-mail: Hartung_v@ukw.de

Photon-counting detector technology has enabled a decisive step forward in recent years. While substantial radiation dose reduction potential in comparison with conventional energy-integrating detectors could be demonstrated for CT examinations[9], only proof-of-concept designs exist for flat detector systems useable in endovascular interventions[10]. Apart from the optimization of existing technology, the greatest radiation-saving potential lies in avoiding ionizing radiation for image acquisition altogether.

Magnetic particle imaging (MPI) is a rapidly evolving technique first introduced in the early 2000s[11]. As a tracer-based tomographic method based on magnetic nanoparticles, it can be used for multiple applications, such as vascular imaging[12,13], oncology[14,15], cell tracking[16], and inducing localized hyperthermia[17,18], among others. Without any background signal and real-time capability[19,20] it is suited for vascular imaging and possibly interventions[21]. However, data regarding in-vivo applications are scarce, since only limited experience exists with pre-clinical scanners and small animal experiments[22,23].

All MPI tracers currently in use contain superparamagnetic particles of iron oxide (SPIOs). Resotran®, a Ferucarbotran-based tracer reminiscent of the discontinued MRI contrast agent Resovist®[24], was recently approved for in-vivo applications. Initial studies have shown that Resotran® is equally suitable as a tracer for MPI as Resovist®[25]. Although this theoretically allows the application in humans, studies reporting on Resotran® as a tracer for MPI-based vascular imaging are still lacking.

Due to the limited field of view and scanner size, possible applications of MPI in humans have mostly focused on neuroimaging[26]. So far, two concepts for brain imaging[27,28] and one concept for perioperative tumor assessment have been realized[29]. The upscaling of MPI hardware to human size for other applications has been restricted by specific absorption rate (SAR) limitations and peripheral nerve stimulation, resulting in large and slow systems with limited flexibility. On track to a stand-alone MPI scanner for vascular interventions, a hybrid concept with partial substitution of X-ray in an angiography lab appears promising. In this regard, a device suitable for clinical application should integrate seamlessly into an existing environment designed for vascular interventions.

While a lightweight and portable human-sized MPI scanner designed for vascular interventions of the lower extremity within a catheter lab was recently introduced, its capabilities have only been demonstrated in a mock phantom thus far[30]. This investigation aims to assess the MPI system's performance in a realistic human cadaveric model with continuous extracorporeal perfusion[31] employing a tracer approved for clinical use in humans.

We demonstrate that the scanner integrates seamlessly into a standard angiography suite and enables real-time, radiation-free visualization of the femoral vasculature. The system reliably detects tracer boluses of 2 ml Perimag® or 1.5 ml Resotran®, with simultaneous imaging using digital subtraction angiography (DSA) confirming its feasibility and reproducibility. These findings highlight the potential of MPI as a viable alternative for guiding peripheral vascular interventions while reducing radiation exposure.

## Methods

### Cadaveric specimen and extracorporeal perfusion
The anatomical institute of our university provided two cadaveric specimens for the presented experiments. The body donors consented to the use of their remains for study and research purposes in their lifetimes. The institutional review board (Medical Ethics Commission of the University of Würzburg) waived the need for approval (protocol number: 20220413 01). Extracorporeal perfusion was established via surgically prepared inguinal and infragenicular arterial accesses and introduction of conventional angiographic introducer sheaths (7F × 13 cm, Flexor©, Cook Medical, USA) in the common femoral artery for inflow and popliteal artery for outflow. The sideports of the introducer sheaths were connected to a reservoir containing a mixture of Ringer's and glucose solution and a reservoir for waste, while a peristaltic pump generated (Ismatec MCP-Z, Cole-

Parmer GmbH, Germany) continuous circulation. A Y-connector was placed between the sideport of the inflow sheath (inguinal) and the perfusion circuit to allow manual contrast injection. A detailed description of the model can be found elsewhere[32]. The success of model establishment was defined as sufficient leak tightness under pump flow rates necessary to allow realistically-appearing flow in DSA. Furthermore, accumulation of contrast agents or tracers in or around the intended field of view should not occur under continuous perfusion. After preparation, the cadaver was transferred to the angiography lab of the radiologic department.

### Scanner setup
**Scanner concept and image generation.** The technical details of the human-sized interventional MPI system are introduced elsewhere[30]. The scanner is based on the traveling wave MPI concept[33] and generates a field-free line (FFL) for spatial encoding, which is steered dynamically along specific trajectories through the field of view using a dedicated electromagnetic coil system. When the FFL is steered over an area consisting of magnetic nanoparticles, the magnetization of the particle ensemble flips, and this rapid change in magnetization is measured inductively as a signal in a receive coil. This signal is co-registered with the known position of the FFL employing an image-based system matrix reconstruction[34], which results in a 2D projection image (in x–z-projection).

**Hardware components.** The experimental assembly consists of a control unit with corresponding input/output periphery, transmit coils ("the scanner") and receive coil. On the transmit side, an arbitrary wave generator generates the designed sequence, which is amplified (AMP-cabinet, Hoellstern, Germany) to realize the high magnetic fields required for encoding and visualization. On the receive side, a flexible receive coil is wrapped around the body part of interest, in this case the upper leg. There, the signal is inductively acquired, filtered, amplified, and digitized using an analog-digital converter (ADC). The received signal is then reconstructed and visualized in real-time on the control unit (Pure Devices GmbH, Germany)[34]. The assembly is outlined in Fig. 1.

### Interventional MPI scanner setup
The interventional MPI scanner is portable and lightweight, consisting of approximately 8 kg copper, 1.2 kg polylactic acid, and 0.8 kg epoxy resin. It was built to fit a human leg and allows a bore size of 20 cm and a length of 30 cm. To generate the FFL, the MPI scanner uses three drive coils. Two overlapping saddle-coil pairs operate in Helmholtz configuration at a frequency of 60 Hz, each, with a phase shift of 90°. A pair of solenoids operating at 2.480 Hz is used to steer the FFL. The coil assembly was designed to provide a radiolucent window for concurrent X-ray operation. At a power consumption of up to 14 kW, the scanner delivers a gradient strength of up to 0.36 T/m (0.25 T/m at 70% system power) and generates drive field amplitudes of up to 70 mT, which result in a SAR of 0.15 W/kg. A schematic of the principal components of the scanner can be found in Fig. 1. An overview of the operating parameters of the MPI scanner and X-ray unit is provided in Table 1.

### Contrast agent and tracer
X-ray contrast agent was Imeron 350 (714 mg Iomeprol/ml, Bracco Imaging, Germany) which was manually diluted at a ratio of up to 1:3 with NaCl and tracer, whereas the exact dilution was chosen at the discretion of the operator and mainly determined by the amount of tracer intended to inject. Contrast bolus injection was done manually through a Y-connector under continuous perfusion using 3 ml or 10 ml syringes. Tracers used for MPI were Perimag® (Perimag® plain, 130 nm hydrodynamic diameter, iron concentration 8.5 mg/ml, Micromod Partikeltechnologie GmbH, Germany) and Resotran® (57.4 nm hydrodynamic diameter, 540 mg/ml Ferucarbotran, iron concentration 28 mg/ml, b.e. imaging GmbH, Germany).

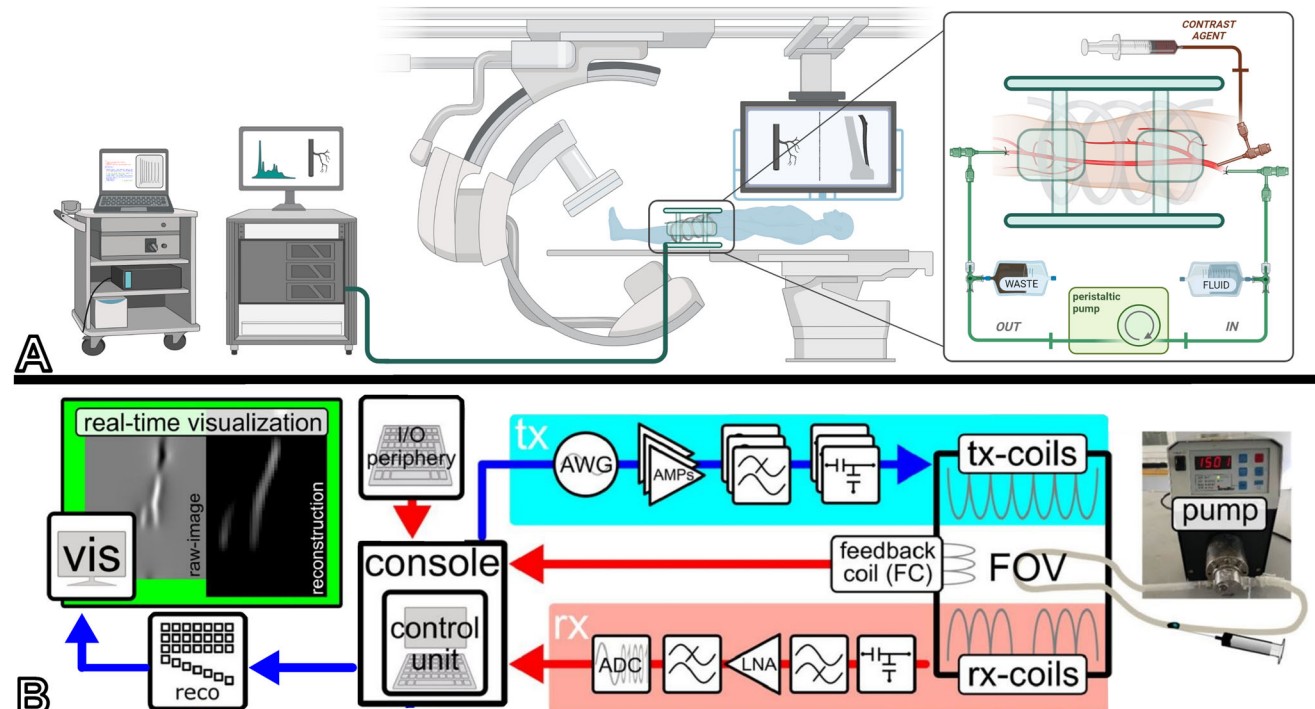

**Fig. 1 | Schematic of the experimental setup. A** Schematic of the arrangement in the angiography lab with I/O periphery and Amplifier cabinet on trays, the ceiling-mounted C-arm X-ray unit, and the patient table with the body donor. The perfusion circuit and MPI scanner are depicted on the right. (Created in BioRender. Gruschwitz, P. (2025) https://BioRender.com/f55o607). **B** Pipeline for image generation: I/O periphery for engaging the scan process, the control unit for activating the transmit chain, drive coils are energized by the AMP cabinet, and signals from the receive chain are then reconstructed and visualized in real-time. Abbreviations: AMP amplifier, I/O periphery input/output periphery, AWG arbitrary wave generator, tx-coils transmit coils, FOV field of view, rx-coils receive coils, LNA low noise amplifier, ADC analog digital converter, reco, reconstruction.

## Experimental assembly

All experiments were conducted in the angiography suite of the radiologic department of a tertiary-care university hospital. Digital subtraction angiographies (DSA) were performed with a ceiling-mounted C-arm system and a floor-mounted patient table (Azurion 7 C20 with 20" flat detector, Philips, Best, The Netherlands). After transferring the prepared cadaver with the installed perfusion circuit to the angiography lab, the cadaver was placed on the patient's table as usual. First, the flexible receive coil was firmly wrapped around the upper leg of the cadaver. Second, the MPI scanner was positioned over the upper leg. Manual cushioning ensured the optimized positioning of the cadaver and the upper leg in the isocenter of the scanner. Due to the scanner design with radiopaque coils placed in an assembly providing a radiolucent window, the flexible C-arm of the angiography unit was placed under fluoroscopy to optimize angulation and maximize the shared field of view. The flat detector of the X-ray system was placed as far away from the MPI scanner as possible to reduce interactions of the scanner with the detector circuitry. Conventional DSA using 5 ml of contrast agent was performed to ensure proper function of the perfusion circuit. As the last step of preparation, the scanner was connected to the amp cabinet and control unit console, which were placed in the back of the angiography suite. A schematic of the experimental setup is shown in Fig. 1.

## Simultaneous MPI and DSA

Upper limits of perfusion flow were confined by the leak tightness of the inflow and outflow sheaths after preparation. Pressure control was not done. A suitable flow rate of the peristaltic pump was determined visually by a physician experienced in angiographic procedures with consecutive DSA series using an iodinated contrast agent until the flow appeared realistic. Afterward, tracers were injected at different amounts and dilutions during MPI to determine a dilution that produced a visually perceivable signal on real-time imaging. Finally, the predetermined amount of tracer and contrast agent was mixed and injected manually under continuous perfusion and simultaneous operation of the X-ray system and MPI scanner.

## Signal-to-noise ratio

After the reconstruction of the data, signal intensity (SI) was calculated as an arbitrary unit. To estimate signal-to-noise ratio (SNR), squared regions of interest with 5 mm edge length were drawn in the proximal ($SI_{prox}$) and distal ($SI_{dist}$) parts of the visualized vessel and in noise ($SI_{noise}$), whereas $SI_{prox}$ and $SI_{dist}$ were averaged to obtain a mean of the temporal maximum SI. SNR was calculated as follows:

$$SNR = \frac{\text{mean maximum SI}}{\text{maximum SI}_{noise}}$$

## Reporting summary

Further information on research design is available in the Nature Portfolio Reporting Summary linked to this article.

## Results

### Experimental assembly

The human body donors enabled a realistic simulation of the examination setup. The interventional MPI scanner could be positioned around the thigh of the body donor as intended. The small size of the required equipment (AMP cabinet, control unit, and wiring) facilitated decentralized positioning in the back of the examination room, which allowed the operator and assisting personnel to perform the angiography in the usual manner. The size of the MPI scanner with the radiolucent window and the receive coils allowed visualization of the mid- to distal third of the superficial femoral artery with an approximate z-coverage of 12 cm and a field of view of approximately 8 cm. The study setting is shown in Fig. 2.

## Table 1 | Imaging parameters

| DSA | | MPI | |
|---|---|---|---|
| Tube voltage [kV] | 68 | Drive coil frequencies [Hz] | 60, 2480 |
| Tube current [mAs] | 2–10 | Drive field amplitude [mT] | 70 |
| Power consumption [kW] | 4.5 | Power consumption [kW] | 14 |
| Image dimensions [cm] | 12 × 8 | Image dimensions [cm] | 12 × 8 |
| Max field of view [cm] | 40 × 30 | Max field of view [cm] | 25 × 12 |
| Image matrix | 2586 × 1904 | Image matrix | 50 × 24 |
| Pixel size [mm²] | 0.154 × 0.154 | Resolution [mm] | 5 |
| Temporal resolution typical/ max [fps] | 1/6 | Temporal resolution typical/ max [fps] | 5/10 |
| Kerma per frame [mGy] | 0.1–0.2 | SAR [W/kg] | 0.15 |

### Perfusion model

The establishment of the continuous perfusion model was successful in all three cadaveric legs. Relevant fluid leaks did not occur when the pump flow rate was increased until flow appeared realistic at DSA. The perfusion circuit was stable and allowed manual injection of a mixture of contrast medium and tracer. No relevant amounts of tracer or contrast agent accumulated in the surrounding soft tissue. The visualization of the femoral arteries by DSA was typical after contrast administration. The superficial femoral artery was patent without perceivable clots. Vascular pathology, such as wall calcification and stenosis, could be assessed as usual. Muscular branches were also patent and perceivable in DSA. Manual measurements established the superficial femoral artery diameter at approximately 5 mm with some variation owing to wall irregularities and measuring error due to projection and focus-object distance (Fig. 3). The examination setting did not interfere with the operator's freedom of movement (Fig. 2, Fig. 3).

### Tracer dilution and pump flow rates

The effective pump flow rate was not measurable. Patent branches and minor leakage at the inflow- and outflow sheaths prevented an objective determination of tracer dilution at the site of scanning. Therefore, the flow rate was measured indirectly. After determining a visually realistic flow using DSA, the outflow was determined over a period of 10 minutes. This resulted in a calculated net flow rate of about 108 ml/min not accounting for losses by capillary leak. Under these circumstances, tracers were injected at the Y-connector on the inflow sheath, which was approximately 23 cm upstream of the isocenter. Tracer dilutions of either Perimag® or Resotran® of 1:10, 1:5, and 1:2 did not yield visually detectable signals on real-time MPI. Therefore, systematic assessment and postprocessing to determine the lowest concentration detectable with the current scanner configuration was omitted. Satisfactory DSA images and MPI signal were achieved by injection of either 2 ml of Perimag® with 1 ml of Imeron 350 or 1.5 ml of Resotran® (1 vial) with 1.5 ml of Imeron 350. A contrast bolus with a tracer dilution of 1:2 or less was not detectable with MPI.

### Imaging concordance, visualization of vasculature, simultaneous X-ray, and interventional MPI operation

The spatial resolution of an MPI scanner depends on several parameters, such as the magnetic gradient strength, the used trajectory and frequencies, and the used particle system, and can be estimated here to be about 5 mm[30,35]. For image reconstruction, the image-based system matrix approach is used[36]. For that, the model-based system matrix is pre-calculated using a virtual MPI scanner and a magnetization model based on the Langevin theory[37]. All data are processed and visualized in real time using a dedicated reconstruction framework[37]. Under the given circumstances, the MPI scanner was able to display the superficial femoral artery in the mid-to-distal third. At typical flow rates and X-ray settings, bolus influx

and clearance took two to three seconds. While this corresponds to three DSA frames, MPI displayed the tracer bolus passage at up to 10 fps (Fig. 3). The low-grade stenoses <50% seen on DSA were not perceivable with MPI. Vascular branches visible with DSA were not perceivable with MPI. The radiopaque coils of the MPI scanner reduced the shared field of view to 12 × 8 cm. The receive coil partially obscured the superficial femoral artery, but this could largely be remediated by digital subtraction. Scanner operation produced substantial acoustic noise. Interactions of MPI and detector circuitry caused characteristic streak artifacts with subsequent minor deterioration of image quality (Fig. 4A).

### Reproducibility of results

The entire examination setup, including cadaver positioning, mounting the interventional MPI scanner, and setting up the equipment, was successful in two cadavers and three upper legs (two left, one right). The parameters chosen for realistic perfusion, tracer-contrast agent dilution, and scanning were transferable among specimens and produced comparable results (Fig. 4).

### Signal-to-noise ratio

Maximum SI for the Perimag® bolus was 332.6 AU for the proximal vessel ROI and 384.1 AU for the distal vessel ROI, respectively. For Resotran®, maximum SI was 152.8 AU proximally and 178 AU distally. Maximum SI for noise was 15.1 AU for both measurements. This resultet in a SNR of 23.7 for Perimag® and 10.5 for Resotran® (Fig. 5).

### Discussion

This study provides a proof-of-concept study of a human-sized MPI scanner for vascular interventions under realistic conditions, which produces images resembling DSA in terms of image contrast and temporal resolution and allows the assessment of femoral vessels. The presented scanner[30] covers a field of view of 12 × 8 cm thereby allowing to visualize the superficial femoral artery. The technical setup is compact enough to be integrated into an existing conventional angiography laboratory. In its current form, the setup may enable hybrid interventions guided by conventional DSA and MPI, which can be considered a major step towards X-ray-free interventional angiography. As of this writing, only three other MPI-based devices designed for application in humans are described in the literature[27–29]. However, the applicability to the human extremities, together with the compact design, renders the presented scanner unique.

The Z-coverage currently realized by our scanner lies well below what is reached under ideal circumstances for DSA. With optimum patient anatomy and geometry of the image generation chain and using the largest detectors commercially available for angiography units (approximately 20 in. diagonal), the usable Z-coverage for DSA can reach up to 30 cm but is usually far lower. However, with the current scanner design and limitations induced by the large bore size, a significant increase in Z-coverage does not seem likely in the near term. Nevertheless, MPI is not restricted by considerations of radiation dose. In turn, repeated imaging to cover the length of an upper leg is much less problematic than for DSA.

The cadaveric perfusion model allowed for a realistic simulation of the examination of the human vascular periphery. DSA of the phantom appeared visually realistic with net flow rates determined at about 108 ml/min. This corresponds well to the normal range seen in healthy individuals with blood flow between 100 and 150 ml/min[38,39]. As shown in previous studies, the model is adaptable to a range of different study approaches and does not interfere with the operator during interventional treatment[9,32]. With this study, we were able to demonstrate that the model can be adopted successfully for parallel imaging of DSA and MPI. As the phantom is suitable for angiographic interventions[32], evaluation of the MPI scanner for image guidance of endovascular procedures will be the topic of further investigations. Angiographic materials such as guidewires, catheters, and angioplasty balloons can be labeled for visualization in MPI[30,40]. Furthermore, it has been shown, that signals from different SPIOs or different binding states of SPIOs in the same volume of interest can be separated,

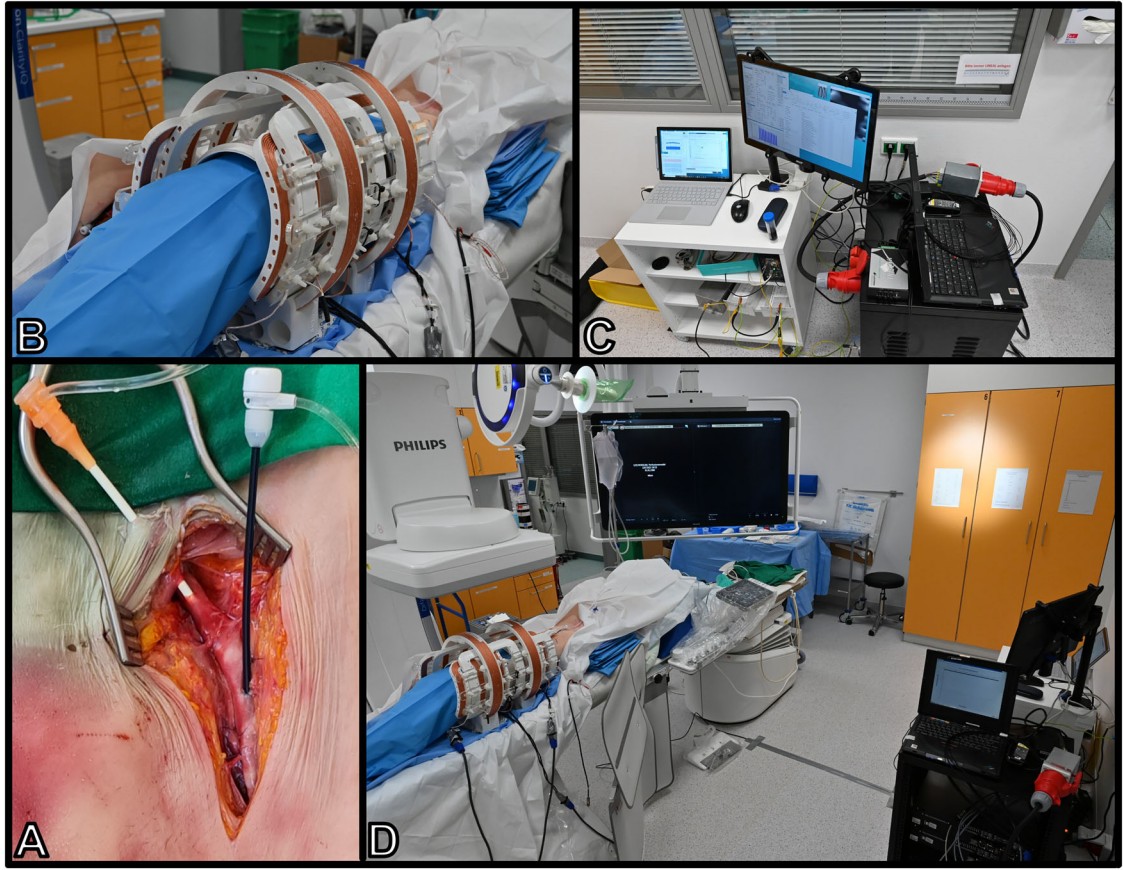

**Fig. 2 | Preparation of the cadaver and setup in the angiography suite. A** Groin after surgical preparation of vascular access. **B** Scanner assembly with transmit and receive coils placed over upper leg (cadaver draped in blue). **C** Control unit, I/O periphery and AMP cabinet in the back of the operator's position. **D** Overview of whole assembly illustrating spacious operating situation. Abbreviations: I/O periphery input/output periphery, AMP amplifier.

which is termed multi-color MPI and seems promising for differentiating labeled instruments from luminal contrast after bolus injection of tracer[41,42]. Consequently, MPI-guided intervention with labeled materials will be the topic of future studies.

We could demonstrate that the MPI scanner produces angiographic images with comparable image quality for two different tracers. While Resotran®, as a contrast medium primarily for MRI has recently been approved for human use in Europe, Perimag® is widely adopted for research purposes but not approved for clinical use as of this writing. Perimag® realizes an SNR roughly twice as high as Resotran®, which is in keeping with previous studies comparing signal amplitude between MPI tracers[43]. Although the currently achievable SNR suggests that further dilution of tracers might be feasible, the tested dilutions were not distinguishable visually in real-time imaging. In this context, it is worth mentioning that many other tracer systems exist, which can be specifically tailored to a clinical need. For angiographic purposes, vastly improved tracers delivering orders of magnitude better signal performance are currently under development[44]. Furthermore, the functionalization of nanoparticle surfaces is an especially successful approach for tracers aimed and biochemical functionality[45,46]. Nevertheless, Resotran® and Perimag® are currently available commercially, and Resotran® is the only tracer available with approval for use in humans.

It can be assumed that Resotran® is well tolerated as it is a re-approval of the MRI contrast agent Resovist®, for which there is plenty of experience in clinical use[24,47]. Typical peripheral interventions require contrast bolus volumes of at least 50–100 ml, which corresponds to 5–20 bolus injections, depending on the contrast dilution and complexity of the procedure. To achieve a sufficiently high signal-to-noise ratio (SNR), the MPI scanner currently requires one vial of Resotran® as a single bolus, which corresponds

to the maximum approved dose. One vial contains 540 mg of Ferucarbotran or 28 mg of iron (0.5 mmol). Preclinical studies concerning the chemically similar Resovist® suggested up to 40 μmol/kg bodyweight as safe[24,48]. According to the current study evidence, the administration of up to 6 boluses in a patient weighing 75 kg would be justifiable. Even if this would not be sufficient for a more complex procedure, it proves that the MPI scanner could be used under the current conditions for guidance of basic interventional procedures in humans. This validates the direction of research and strongly encourages further improvements in the design, especially with regard to sensitivity.

The current spatial resolution of 5 mm corresponds to the average diameter of the whole SFA. This value has been established in laboratory tests beforehand but could not be systematically investigated in the current setting. An exact determination of the discriminatory resolution was hence not available. Spatial resolutions achievable with preclinical MPI scanner have been reported at up to 1 mm with the main determinants being bore size and magnetic gradient strength[36,49,50]. In the context of human applications, the particle systems are limited to Resotran®. Gradient strength is limited by the technical prerequisites of the scanner design and safety margins concerning SAR and peripheral nerve stimulation. The other aspects will certainly be addressed in the next evolutionary step of the scanner design.

Whereas clinically significant stenosis is usually determined at >60% vascular narrowing, the body donors examined displayed only minor stenoses of <50% at a vessel diameter of 5 mm, which could not be resolved. However, previous vessel phantom studies demonstrated that clinically relevant stenoses >70% are resolvable[30]. Furthermore, in clinical scenarios, when severe calcification obstructs sufficient visual assessment of stenosis on DSA, obstructed flow or, rather, delayed bolus passage is indicative of

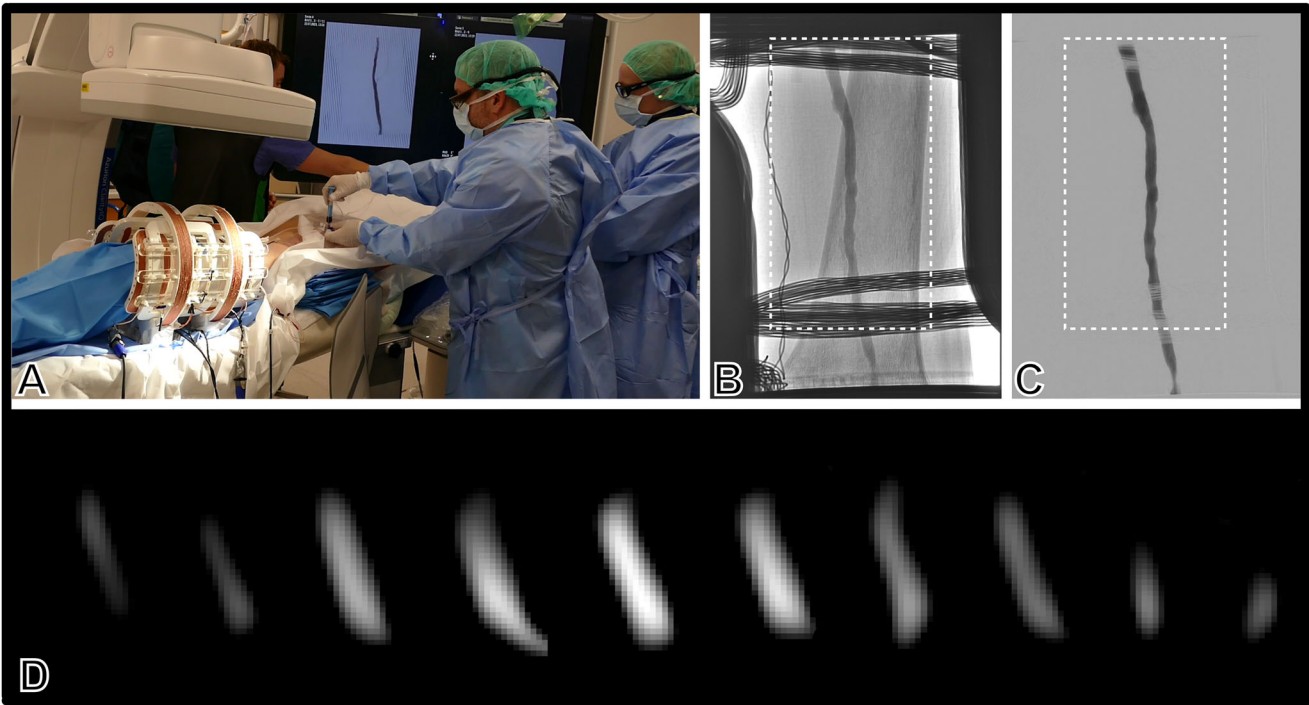

**Fig. 3 | Interventional guidance via simultaneous digital subtraction angiography and magnetic particle imaging. A** Operators injecting a tracer-contrast agent mixture to acquire DSA and MPI images simultaneously (images reproduced with permission of all depicted individuals). **B** Angiogram without subtraction displaying the shared field of view of DSA and MPI (12 cm × 8 cm); note the X-ray window confined by the scanner and overlaying receive coils. **C** Digital subtraction angiography with realistic appearance of the perfused superficial femoral artery at 1 fps. **D** Frame series from 2 seconds of MPI at 5 fps after injection of 1.5 ml Resotran® with 1.5 ml of iodine contrast agent. The contrast bolus influx and clearance are accurately visualized. Note: Dashed frames on DSA indicate the shared field of view (appr. 12 × 8 cm). Abbreviations: DSA digital subtraction angiography, MPI magnetic particle imaging.

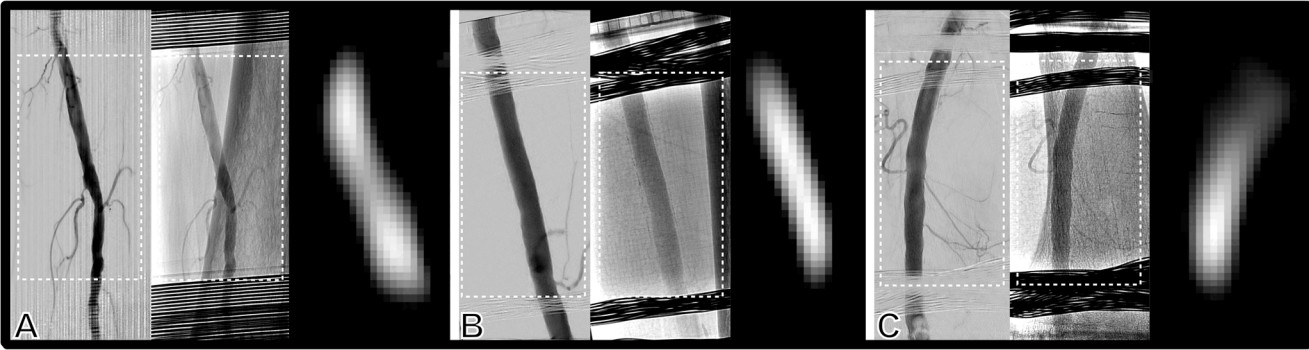

**Fig. 4 | Reproducibility of results.** Angiogram with (left) and without (middle) subtraction and corresponding MPI frame (right) of the shared field of view acquired in two left legs (**A**, **B**) and one right leg (**C**) from two human cadavers. Note the obscuring effect of receive coils (**A**–**C**) and streak artifacts (**A**) with minor impairment of DSA image quality. Note: Dashed frames on DSA indicate the shared field of view (appr. 12 × 8 cm). Abbreviations: DSA digital subtraction angiography, MPI magnetic particle imaging.

significant stenosis. The high temporal resolution leads us to hypothesize, that a clinically significant stenosis could be detectable with the current setup, either in terms of visually perceivable delay of bolus passage or in terms of signal strength or tracer concentration in the stenosis[51]. Unfortunately, body donors are difficult to pre-select, so the combination of patent vessels for continuous perfusion and higher-grade stenoses is rarely encountered. With this study focusing on the visual assessment of bolus passage (as in DSA), another important aspect of MPI is the potential to quantify the detected signal[51]. Quantification combined with the high temporal resolution of the presented MPI scanner might enable indirect stenosis detection utilizing the resultant change in flow dynamics. Flat-detector angiography is limited to assessing flow changes caused by dissections or heavy calcifications. Although an increase to 6 fps is technically feasible, this leads to a substantial increase in radiation exposure. In contrast, interventional MPI guidance allows for superior temporal resolution beyond the current 5 fps without any dose burden. This fact supports the complementary nature and opportunities of hybrid imaging. Further investigations are required to exploit the potential of flow quantification in combination with high spatial resolution.

Despite very high liver first pass, SPIOs recirculate to significant amounts, which translates to relevant SPIO concentrations in the venous system. This leads to venous contamination with the potential to impede arterial angiography on T1-weighted MRI[52]. Theoretically, this would affect MPI angiography as well, especially at the current spatial resolution. As a limitation of our study, the perfusion model is unsuitable to display venous return. However, we do not consider venous contamination a relevant issue.

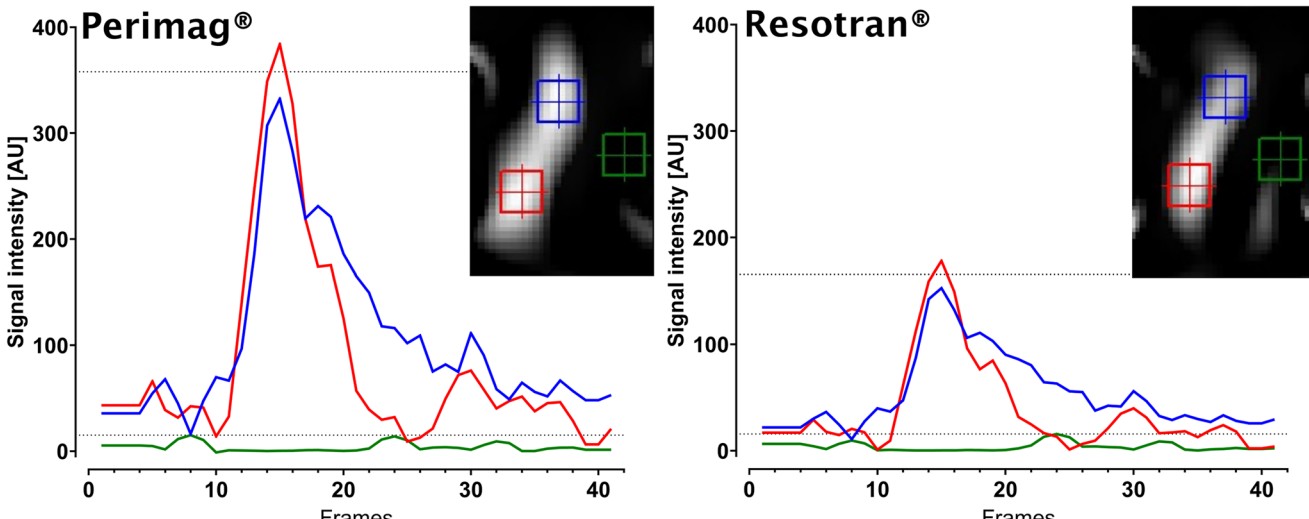

**Fig. 5 | Signal-to-noise ratio of Perimag® and Resotran®.** Signal intensity (SI) measurements from contrast bolus passage in the mid-superficial femoral artery at a temporal resolution of 200 ms (5 frames/s). SI measurements from ROIs in the proximal (blue) and distal (red) vessels and from noise (green). Corresponding MPI angiography frames in the upper right corner. Horizontal dotted lines indicate maximum SI from noise and mean maximum SI, respectively. SNR was determined at 23.7 for Perimag® and 10.5 for Resotran®. Abbreviations: SNR signal-to-noise ratio. Note: The field of view is approximately 12 × 8 cm.

First, there is the aspect of dilution. Venous return from capillaries significantly reduces tracer concentration and even more so after the liver's first pass, which is roughly 80%, and reaching systemic circulation with concurrent dilution in 6 l of blood volume for the average person. Second, the venous return of a contrast bolus can be clearly differentiated owing to slower flow, typical anatomy of the veins, larger diameters, and bulbous valves. Third, MPI is limited in its dynamic range to render vastly differing concentrations of tracer next to each other[53]. Whereas this is considered a problem in many applications, it allows the suppression of venous return, thus eliminating contamination.

The scanner assembly delivered a SAR of 0.15 W/kg, which is well below the normal limits established by the WHO and IEC at 2 W/kg (whole-body) or even 20 W/kg for the extremities. Furthermore, the scanner yields a maximum peak-to-peak amplitude of 70 mT. This is below the limit for peripheral nerve stimulation, which was determined in advance to be 80 mT for the current hardware configuration and settings[30]. The acoustic noise nuisance from the scanner is substantial due to the lack of shielding, which currently leads to both an unpleasant environment for a potential patient and restrictions in communication between the operators and from operator to patient. Another considerable safety issue could stem from metallic implants, with possible effects being signal interference with the scanner itself, heating, damage to the circuitry of pacemakers, or hazardous power induction. We conducted experiments with metallic hip implants without significant heating or interference and further systematic studies with pacemaker devices and leads are underway. Others reported substantial heating of some stents but not for models intended for use in the peripheral vasculature[54,55]. At the current stage of development, the devices are neither equipped for sterile use nor sufficiently resistant to moisture.

According to the results presented above, we identified specific fields of improvement. First, integration of the MPI components into the angiography lab appliances is necessary. The MPI scanner meets the requirements to be designed as a removable add-on solution with connectors integrated into the patient table reminiscent of MRI coils. Input/Output periphery and control units could be integrated into the preexisting periphery (foot-operated switch, monitor, etc.), as is already the case with add-on modalities, such as intravascular ultrasound or vital stats monitoring. Power generation components would fit into the control cabinet housing of the X-ray generator and switchboards.

Second, shielding is paramount. For the transmit chain, electromagnetic shielding would greatly reduce interference with X-ray image generation and acoustic shielding would reduce noise, thereby ensuring unrestricted communication of patients and operators during the procedure. Improved shielding of the receive chain would reduce signal noise and increase SNR. Third, improvements in postprocessing algorithms, including denoising and image optimization, are necessary. Fourth, improved designs of flexible receive coils are desirable to allow adaptation to a wide range of patient constitutions and improve SNR. Fifth, increasing power supply with concurrent measures for heat dissipation and the respective high current resistant circuitry would significantly increase SNR as well as temporal and spatial resolution[30].

At the current stage of development, MPI is not fully capable as a standalone modality owing to obvious confinements in the field of view and spatial resolution. However, with radiation dose reduction in mind, in a hybrid approach, the MPI system could replace the most radiation-intense parts of an endovascular procedure, which are the DSA series. These are usually only necessary to diagnose disease and verify the successful intervention. All other parts, such as guidewire probing and device placement, which are usually done under fluoroscopy with much-reduced radiation exposure, could still be done under X-ray guidance. Thus, with the current stage of evolution, MPI would already provide a significant added value to radiation dose reduction. Based on personal experience, approximately 80% of the total radiation exposure in peripheral interventions is contributed by the DSA series, and only 20% stems from fluoroscopy. Both parts could be potentially substituted in part by MPI. As a rough estimation, it can be assumed that MPI, once fully developed and integrated, could realize a dose reduction of up to 70%. However, further systematic studies are necessary to determine the extent of radiation dose reduction by substituting the DSA series. The assessment of cost-effectiveness could substantially for other regions of the globe, depending on the importance of radiation protection demands.

There are several limitations to the currently available tracer Resotran®. One is the low quantity allowed for injection under the current approval. There are several other tracers in development[36], of which some have shown substantially higher SNR[44]. Since the development and approval of new tracers require high amounts of time and financial resources, a corresponding demand must be generated by available and clinically relevant technology.

Several limitations have to be acknowledged. First, since this is an experimental proof of concept study investigating two body donors, the results may not be transferable or generalizable. Second, the current assembly offers a limited field of view to a small segment of the superficial femoral artery. Third, the perfusion model does not offer a venous return of contrast, which reduces realism. Fourth, the currently achievable spatial resolution can be assumed to allow the detection of high-grade stenoses although this cannot be proven due to the lack of appropriate lesions. A suitable resolution phantom for testing and validation is currently under development and will be available soon to specify and compare the performance of MPI scanner assemblies. Fifth, the single tracer that is currently clinically approved allows only one injection in the required quantity according to the maximum approved dose. Sixth, the approved route of access for Resotran is intravenous injection. Although this allows some inference about the safety profile, intraarterial injection of Resotran cannot self-evidently be considered safe, and further studies are needed to approve this route of access. Seventh, in parallel operation of the MPI scanner and DSA, the image quality of the latter is impaired by interactions with the image processing system and resulting streak artifacts, as well as opacification from the MPI scanner. Eighth, systematic control or tracking of flow, pressure, or temperature was not done.

In conclusion, this study inaugurates a human-sized MPI scanner intended for vascular interventions that operate in a realistic environment using an extracorporeally perfused human cadaveric model. It enables visualization of the femoral vasculature with a clinically approved tracer. The experimental design was reliable and delivered reproducible results. The high spatial resolution of DSA and high temporal resolution of MPI complement each other for improved vascular assessment. With the dimensions of the scanner assembly, MPI reaches a milestone on the way to a fully MPI-based procedure by allowing an intermediate stage of hybrid interventions with DSA and MPI guidance for potential radiation dose savings.

## Data availability

The datasets generated and/or analyzed during the current study are not publicly available but are available from the corresponding author on reasonable request. Due to the nature of this research, participants of this study did not agree for their data to be shared in a public repository.

## Abbreviations

| | |
|---|---|
| DSA | digital subtraction angiography |
| FFL | field-free line |
| MPI | magnetic particle imaging |
| SAR | specific absorption rate |
| SNR | signal-to-noise ratio |
| SPIO | superparamagnetic particles of iron oxide |

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

## Author contributions

V.H. and P.V. designed and supervised the study analyzed all data, and prepared the paper. V.H., P.G., A.M.A., J.P.G., J.G., T.R., T.K., and P.V. conducted the experiments. A.M.A., J.G., T.R., and T.K. supported the preparation of the paper and figures. P.G. and J.P.G. revised the paper. S.H. and T.A.B. contributed to the preparation of the paper and provided quality control. D.P. prepared the cadaveric specimen. S.E., F.K., M.A.R., V.C.B., and T.A.B. provided material support. All authors read and approved the final paper.

## Funding

## Competing interests

Patrick Vogel is employed by Pure Devices GmbH (Rimpar, Germany). The remaining authors declare no competing interests.
