## [Transparent Peer Review file · Communications Medicine]

Magnetic particle imaging angiography of the femoral artery in a human cadaveric perfusion model

Corresponding Author: Dr Viktor Hartung

Version 0:

Reviewer comments:

Reviewer #1

(Remarks to the Author)

This manuscript reports a first-of-its-kind demonstration of magnetic particle imaging (MPI) angiography of the femoral artery using a human cadaveric perfusion model. It is a demonstration of the use of a previously reported interventional MPI (iMPI) scanner that can be used in an interventional suite. The potential use is to replace the use of X ray and CT imaging for patients with peripheral arterial disease. The work appears well executed and the results are presented well, along with an exhaustive discussion of their limitations. Although the work does not conclusively demonstrate superior performance of MPI over digital subtraction angiography (DSA), the work reported in the manuscript is impactful as the first demonstration of successful tracking of a bolus of MPI tracer in a human femoral artery. Furthermore, the authors discuss many of the technical and logistical challenges that must be overcome prior to using the iMPI scanner in living patients. As such, I think this manuscript will serve as a milestone in MPI development and will motivate further research and development to overcome the challenges identified by the authors. My only recommendation is that the manuscript be edited as there are several spelling mistakes throughout.

Reviewer #2

(Remarks to the Author)

In this paper, the authors utilize human-sized MPI scanner facilitates reliable radiation-free image guidance for peripheral vascular interventions in the superficial femoral artery with a tracer approved for use in humans. The overall is meaningful enough to meet the journal now, thus I suggest the following minor points after careful assessments.

1. The authors offered the usage volume of the tracers (Perimag, Imeron, Resotran), what is the concentration of these tracers?
2. The main of Perimag or Resotran are nano Fe₃O₄, how long residence time in blood vessels? Weather the long residence time would affect the DSA imaging and therapy?
3. The authors claimed that tracer dilutions of either Perimag or Resotran of 1:10, 1:5 and 1:2 did not yield visually detectable signal on real-time MPI. Weather the usage of the tracers was too much for causing long-term safety concerns?
4. Recently, lots of papers about vascular interventions by MPI have been reported, please elaborate the unique advantages in peripheral vascular interventions in the superficial femoral artery.
5. Although the tracers (Perimag and Resotran) used for MPI and Imeron 350 for CT are safety, the safety of the mixture used for MPI/CT imaging should be confirmed.
6. The result is indeed meaningful. The limitations claimed by the authors is very important to be proved and be mor researched.
7. The authors should discuss the other magnetic tracers (ACS Nano 2023, 17, 14, 13792–13810; ACS Nano 2023, 17, 10, 9529–9542; Science Bulletin 2024, 69, 5, 636-647; Chem, 2022, 1956-1981; Angew. Chem. Int. Ed. 2022,61, e2021172; Nature Biomedical Engineering, 2020, 4, 325–334) would obtain the better imaging performance.

Reviewer #3

(Remarks to the Author)

3109 Review, Magnetic particle imaging angiography of the femoral artery in a human cadaveric perfusion model

Brief Summary of Manuscript

This work is about illustrating the potential clinical usability of magnetic particle imaging in conjunction with digital

subtraction angiography for peripheral artery disease. In terms of development spectrum, it is proof of concept, but the fact that it is very focused on usability in a realistic clinical setting is important.

Overall impression of the work

From a technical perspective, ranked on low-medium-high, the technical advancement of MPI through this work is low to medium. From a usability perspective, ranked on low-medium-high, the clinical usability advancement of MPI through this work is medium to high. It is solid enough work to warrant publication, following revisions. In some places it overstates the advancement and/or conclusions and sometimes leaves out important clinical context. All of which is likely addressable through revisions.

Specific Comments

1. Minor, line 65: Most tracers currently in use contain superparamagnetic particles of iron oxide (SPIOs). This sentence reads strangely ... recommend "Most MPI tracers currently in use ..."

2. Minor, line 77: Referring to Resovist and Resotran – It would strengthen the presented work to note that the approved ROA (route of administration) is similar to what was used in this work; approved ROA is IV, in this work arterial injection (these are two different ROAs, but more similar than say IV and subQ). The point is that these ROA's have higher safety risks, as compared to say IOs approved for subcutaneous injections (e.g., Magtrace), so having an approved tracer with the higher risk ROA is important.

3. Either in Results or Methods section, can you provide sex and age of cadaveric donors? Understandable if this is not possible because you are blinded to this.

4. Line 96, although you provide Table 1 comparing DSA and MPI, can you additionally provide the typical needed FOV (particularly in z, assuming z is along the length of the leg) for PAD DSA?

5. Perfusion model: Was there pressure control? Was there temperature control? The former being more important in an ex vivo setting like this. Not a show-stopper if the answer is 'no', but should be explicit and transparent about having/not having both. In future work, should be able to control and/or monitor both in either phantoms or cadaveric phantoms like this.

Comments 6-8 are closely related.

6. Line 99: "The establishment of the continuous perfusion model was successful in all three cadaveric legs." How was success determined? i.e. based on what metric? Pressure measurements of some sort? Visualizing flow down to a certain level? Other? You may allude to this in the methods section – clinician experienced with visualizing DSA in PAD. Thank you for including in the Results section to..

7. Lines 403-405: A bit out of order, but related to comment just above. "A suitable flowrate of the peristaltic pump was determined visually by a physician experienced in angiographic procedures with consecutive DSA series using iodinated contrast agent until flow appeared realistic." Interesting metric. Not a bad idea. Kind of realistic, but may also be misleading depending on 'resolution' of DSA with respect to flow rates.

8. Line 110. Effective pump flow rate was not measurable. In future work, consider using Transonic flow probes. They have nice in line flow probes or wrap around (vessel or tubing).

9. Was flow steady or pulsatile? If pulsatile, close to human heart rate? Pointing this out in terms of if pulsatile flow impacts DSA (likely minimal) and/or MPI.

10. Line 128, is the 5 mm resolution isotropic?

11. Line 129, need a period after used and ref 33 and before For

DISCUSSION

12. Overall: Good that basic SNR measurements were provided. But what about geometric measurements? those are the most important, along with functional measurements like flow rate (e.g., typically by Doppler), when deciding on when to intervene for PAD as well as whether the interventional procedure has successfully re-opened the vessel. Do you currently have the resolution for any of this?

13. Related note: Ah, the main point of this work is what is done once the decision to intervene has been made. So, how about adding in the Intro a brief description of the clinical work up prior to DSA and revascularization, e.g. often Ultrasound (geometric and doppler). This would help further define where MPI fits in.

14. Lines 157-9: This study demonstrates the usability of a human-sized MPI scanner for vascular interventions under realistic conditions, which produces images resembling DSA to assess femoral vessels. Please consider: "This study provides a proof of concept usability study of a human-sized ...". Usability studies are so important for eventual clinical adoption. Too many times scientists and engineers develop stuff that clinicians or the workflow itself just cannot use. But a true and thorough usability study does go beyond what is presented here.

Regarding 'images resembling DSA' ... a little of an overstatement? Starting with Figure 4 ... are the MPI images the same

length as displayed in the concomitant DSA? If not, please put a red dotted line (or some other demarcation around the DSA that the MPI is representing. If so, resembling is a bit of an over statement. Could you provide an order of magnitude for 'resembling' for important metrics? e.g. FOV, scan time, recon time, resolution. This could be rough quantifiable estimate, or even a qualitative assessment. But some sort of qualifier seems more appropriate. It is appreciated that the authors started with the term 'resembling' (vs. something more bombastic).

15. Lines 178-183, labeled is mis-spelled a couple times.

16. Line 184: "diagnostic images"

Similar to comment above about images resembling DSA. This appears to be a vast overstatement, at this time. The current figures qualitatively do not look like they would be diagnostic needs, and there are no measurements such as geometry or flow. One way to satisfy this in the future could be to get feedback from the 'customer', i.e. interventional radiologist. For now, please soften this claim.

17. Lines 187-88: Please comment on what causes the difference in SNR for Resotran vs Perimag? Solely based on size? Does one IO clump? Clear faster? What about the IO characteristics and/or interaction with mag fields of MPI result in not insignificant differences in SNR?

18. Line 190, extra comma after suggests and before that.

19. Lines 192-206: So would it be up to 6 vials of Resotran, if one vial = single bolus? please clarify. Also comment on potential adverse events due to iron overload if this, or another clinically relevant administration regime, were to be used.

20. Line 206: You mention sensitivity ... but what about resolution? From the figures comparing DSA to MPI, it appears resolution would be an equal or greater limiting factor right now. And is it easier/harder to overcome resolution challenges, if they are ~equal to sensitivity challenges?

21. Lines 209-210: "An exact determination of the discriminatory resolution was hence not available."

Why not? Couldn't you use the same DSA/MPI set up and use resolution phantoms? Dorenzo not ideal, something more like an artery so something cylindrical with different diameters. An extra experiment is not necessary, per se, because the images in the figures do give the reader a clear sense of differences in spatial resolution between DSA and MPI, but this sort of resolution experimental set up is important and likely to be used multiple times throughout the further development of this MPI application.

22. Line 210: spelling error, should be discriminatory

23. Regarding higher temporal resolution of MPI: This is really a great benefit. Can you please briefly add some commentary on how it is possible the MPI is faster than the DSA? Even with an extensive background in engineering and imaging science, across multiple modalities and preclinical and clinical, I do not intuitively know what the primary factors are here. Is it time to acquire x-rays, for example? Reconstruction? Other?

24. Line 256. Please add acoustic in front of noise here, to explicitly differentiate from signal/image noise.

25. Line 286: spelling error, should be spatial

26. Lines 287-88: But is it cost effective to layer on MPI? If so, to get rid of how much radiation dose? If you happen to have any information about these considerations, please consider adding here. But definitely needed for future immediate next steps in this argument and work.

27. Lines 292-3: Related to above, have you done rough calculations on % reduction of radiation possible, with the use of MPI instead (either full replacement, or as you also highlight partial replacement of DSA procedures)?

28. Line 298: "tracers in development". Again, just a note, have to be careful here. ROA is critical. IV administration has way more risks than subq. It is good that there are, once again, approved IOs that have ROA of venous. Going straight arterial will likely require additional testing.

29. Line 358: Than should be then

30. Line 415: mean maximum SI

31. What is mean maximum? Were there multiple maximum SI measurements made and they were averaged?

32. Line 615: Spelling error should be dotted

Version 1:

Reviewer comments:

Reviewer #2

(Remarks to the Author)

The revised manuscript has addressed our concerns and we have no further comments for author.

Magnetic particle imaging angiography of the femoral artery in a human cadaveric perfusion model

Response to Reviewers

The authors would like to thank all of the reviewers for their valuable feedback on our manuscript. Your remarks are greatly appreciated as they help us to improve the quality of our work. We tried to thoroughly address all of your comments in the following point-by-point response and appropriate modifications have been implemented in the manuscript. Hopefully, we were able to overcome your initial concerns in order to meet the requirements for further consideration of our study.

Reviewer #1

This manuscript reports a first-of-its-kind demonstration of magnetic particle imaging (MPI) angiography of the femoral artery using a human cadaveric perfusion model. It is a demonstration of the use of a previously reported interventional MPI (iMPI) scanner that can be used in an interventional suite. The potential use is to replace the use of X ray and CT imaging for patients with peripheral arterial disease. The work appears well executed and the results are presented well, along with an exhaustive discussion of their limitations. Although the work does not conclusively demonstrate superior performance of MPI over digital subtraction angiography (DSA), the work reported in the manuscript is impactful as the first demonstration of successful tracking of a bolus

of MPI tracer in a human femoral artery. Furthermore, the authors discuss many of the technical and logistical challenges that must be overcome prior to using the iMPI scanner in living patients. As such, I think this manuscript will serve as a milestone in MPI development and will motivate further research and development to overcome the challenges identified by the authors. My only recommendation is that the manuscript be edited as there are several spelling mistakes throughout.

Thank you for the detailed feedback. Your helpful comments and acknowledgement of our work are much appreciated. We revised the entire manuscript for typographical errors.

Reviewer #2

In this paper, the authors utilize human-sized MPI scanner facilitates reliable radiation-free image guidance for peripheral vascular interventions in the superficial femoral artery with a tracer approved for use in humans. The overall is meaningful enough to meet the journal now, thus I suggest the following minor points after careful assessments.

Thank you for acknowledging the importance of our work and for considering it for publication.

1. The authors offered the usage volume of the tracers (Perimag, Imeron, Resotran), what is the concentration of these tracers?

Concentrations, as stated by the producers, were as follows: Perimag has an iron concentration of 8.5 mg/ml, Resotran has an iron concentration of 28 mg/ml and Imeron has an iodine concentration of 350 mg/ml. None of these were diluted but MPI tracers Perimag and Resotran were mixed 1:1 with Imeron before injection.

Unfortunately, realistic concentrations at the site of measurement cannot be reliably estimated.

2. The main of Perimag or Resotran are nano Fe₃O₄, how long residence time in blood vessels? Weather the long residence time would affect the DSA imaging and therapy?

According to the initial Phase II trials for Resovist, which is chemically similar to Resotran^{1,2}, plasma half-life in humans is very short at approximately 8 minutes due to very high liver first pass of 80 % (Kopp et al, 1997). After systemic recirculation and dilution in the blood pool of approximately 6 liters for the average human, no detectable signal would remain. Concerning direct venous contamination, our data indicates that after passage of the capillary bed, Resotran would be sufficiently diluted, so that venous return of tracer would not interfere with signal in the arteries but we could not study this in the presented setting. Iron has a very low X-ray absorption and does not impair DSA³.

3. The authors claimed that tracer dilutions of either Perimag or Resotran of 1:10, 1:5 and 1:2 did not yield visually detectable signal on real-time MPI. Weather the usage of the tracers was too much for causing long-term safety concerns?

The administered amount of one vial of Resotran (= 1.5 ml) is the amount that is considered safe according to its approval. The authors believe that further dilution is unlikely to cause harm. Even five times the approved amount were considered safe in the approval studies but they were considered unnecessary for MRI examinations of the liver and hence not approved⁴. On the other hand, for MPI and the use of Resotran in the context of vascular interventions, it would be desirable to have a safety margin

allowing the injection of approximately 20 boluses or more for a typical intervention in the SFA. Future studies will aim at increasing sensitivity, so that further tracer dilution and hence more complex interventions become feasible.

4. Recently, lots of papers about vascular interventions by MPI have been reported, please elaborate the unique advantages in peripheral vascular interventions in the superficial femoral artery.

Thank you for this very important remark. As stated in the introduction, we cannot stress enough the importance of endovascular procedures in the SFA in the context of peripheral vascular disease owing to the very high demand and also the number of procedures that is currently performed annually. These numbers are likely to increase in the near future in the ongoing trend towards more minimally invasive procedures and improving technical advances in the field. Furthermore, it must be stressed, that, to date, no other scanners exist that provide a large enough bore size to accommodate a human leg.

5. Although the tracers (Perimag and Resotran) used for MPI and Imeron 350 for CT are safety, the safety of the mixture used for MPI/CT imaging should be confirmed.

The authors thank the reviewer for this important remark and fully agree, that safety of the mixture injection cannot be inferred from the data available. Further studies are necessary to confirm safe application in humans.

6. The result is indeed meaningful. The limitations claimed by the authors is very important to be proved and be mor researched.

We would like to cordially thank the reviewer for acknowledging our achievement in this particular case.

7. The authors should discuss the other magnetic tracers (ACS Nano 2023, 17, 14, 13792–13810; ACS Nano 2023, 17, 10, 9529–9542; Science Bulletin 2024, 69, 5, 636-647; Chem, 2022, 1956-1981; Angew. Chem. Int. Ed. 2022,61, e2021172; Nature Biomedical Engineering, 2020, 4, 325–334) would obtain the better imaging performance.

The authors share the reviewer's enthusiasm about the development of tracers specifically aimed at a certain application. Several interesting ways exist, e.g., by improving the synthetization process to increase the magnetization of the particles and furthermore how the surface of particles can be functionalized. This aspect has been added in the discussion and we included the references of Song et al (2020)⁵ and Cutler et al (2010)⁶. In the context of MPI angiography, other tracers exist, which deliver higher performance (i.e. in Tay et al (2021)⁷). However, these are not currently available commercially and only Resotran is approved for use in humans. The authors feel that the other aspects indicated in the references suggested by the reviewer would not add to the principle assertions of the manuscript.

Reviewer #3

Brief Summary of Manuscript:

This work is about illustrating the potential clinical usability of magnetic particle imaging in conjunction with digital subtraction angiography for peripheral artery disease. In

terms of development spectrum, it is proof of concept, but the fact that it is very focused on usability in a realistic clinical setting is important.

Overall impression of the work:

From a technical perspective, ranked on low-medium-high, the technical advancement of MPI through this work is low to medium. From a usability perspective, ranked on low-medium-high, the clinical usability advancement of MPI through this work is medium to high. It is solid enough work to warrant publication, following revisions. In some places it overstates the advancement and/or conclusions and sometimes leaves out important clinical context. All of which is likely addressable through revisions.

The authors would like to thank the reviewers for the valuable feedback and the great remarks, which we feel will enhance the quality of our work.

Specific Comments

1. Minor, line 65: Most tracers currently in use contain superparamagnetic particles of iron oxide (SPIOs).

This sentence reads strangely ... recommend “Most MPI tracers currently in use ...”

all

We amended the manuscript accordingly.

2. Minor, line 77: Referring to Resovist and Resotran – It would strengthen the presented work to note that the approved ROA (route of administration) is similar to what was used in this work; approved ROA is IV, in this work arterial injection (these

are two different ROAs, but more similar than say IV and subQ). The point is that these ROA's have higher safety risks, as compared to say IOs approved for subcutaneous injections (e.g., Magtrace), so having an approved tracer with the higher risk ROA is important. Future application for dsa ... roa i.a. not approved, might carry higher risk for patients → limitations section

We want to thank the reviewer for this important notion and we have augmented the limitations section of the manuscript accordingly.

3. Either in Results or Methods section, can you provide sex and age of cadaveric donors? Understandable if this is not possible because you are blinded to this.

Unfortunately, the authors of this manuscript are not entitled to disclose this data according to applicable laws and regulations.

4. Line 96, although you provide Table 1 comparing DSA and MPI, can you additionally provide the typical needed FOV (particularly in z, assuming z is along the length of the leg) for PAD DSA?

The authors acknowledge, that such a notion would be of great interest to readers in this context. However, it must be noted, that Z-coverage along the long axis of the leg depends on a significant number of conditions. There is the detector size, which depends on the specific equipment and facilities of the respective institution. In the lab, where the experiments were carried out, the X-ray detector has a diagonal of 20 inch. 12-inch detectors or less are in common use elsewhere. Furthermore, the geometry of the image generation chain varies significantly. Important factors are collimation, magnification, focus-detector-distance, focus-object-distance or post-

processing. Whereas in ideal circumstances, a Z-coverage of 30 cm can be achieved in our lab at least for some patients, other circumstances and detector designs limit usable Z-coverage even below 10 cm (i.e. in labs specialized at coronary interventions). To further complement the manuscript, we updated the discussion section accordingly.

5. Perfusion model: Was there pressure control? Was there temperature control? The former being more important in an ex vivo setting like this. Not a show-stopper if the answer is 'no', but should be explicit and transparent about having/not having both. In future work, should be able to control and/or monitor both in either phantoms or cadaveric phantoms like this.

No systematic control of flow, pressure or temperature was available for this proof-of-concept study. Pressure was limited by leak-tightness, that could be achieved at the inflow and outflow sheaths with surgical preparation, which was well below the pressure limitations of the pump. Perfusion fluid was warmed to body temperature before infusion, but significant cooling occurred while circulation through the thawing cadaver. The fresh-frozen cadavers were delivered 24 hours before the experiments and thawed at room temperature. Realistic estimations of core temperature are not available. To further complement the manuscript, we updated the discussion section accordingly.

6. Line 99: "The establishment of the continuous perfusion model was successful in all three cadaveric legs."

How was success determined? i.e. based on what metric? Pressure measurements of some sort? Visualizing flow down to a certain level?

Other? You may allude to this in the methods section – clinician experienced with visualizing DSA in PAD. Thank you for including in the Results section to..methods:

The authors considered success in this context as leak tightness at inflow and outflow sheaths allowing continuous perfusion without accumulation of perfusion fluid, contrast agent or tracer in or at the surrounding areas of the field of view. Pressure was not systematically tracked but typical pump flow rates necessary for realistically appearing DSA were known from previous studies and could be reached after preparation. Flow visualization up to a certain level was no relevant objective because typical SFA diameters in the studied segment are rather uniform and perfusion or patency of smaller branches was not considered relevant in this setting. The methods and results sections have been updated according to the reviewer's suggestions.

7. Lines 403-405: A bit out of order, but related to comment just above. "A suitable flowrate of the peristaltic pump was determined visually by a physician experienced in angiographic procedures with consecutive DSA series using iodinated contrast agent until flow appeared realistic."

Interesting metric. Not a bad idea. Kind of realistic, but may also be misleading depending on 'resolution' of DSA with respect to flow rates.

The authors consider the remarks of the reviewer very important and want to add the information, that three of the authors are radiologists experienced in endovascular procedures. Therefore, these properties could be tuned easily and comfortably to liking. The authors believe, that the reviewer was referring to temporal resolution of DSA being able to visualize the bolus passage appropriately. In this context, typical

framerates of DSA are set at one to two images per second, which corresponds to one to two images needed for the bolus to travel a distance of roughly 25 cm (approximated effective Z-coverage of the X-ray detector). Owing to the high spatial resolution of DSA, this is usually sufficient to assess peripheral arterial disease qualitatively in terms of stenosis characterization and it allows to detect high grade flow reduction. Higher framerates would allow quantitative assessment of flow reduction at the cost of increased radiation dose, which is usually not justifiable. However, the high framerates possible with MPI without the radiation dose burden allow for assessment of stenosis using quantitative measurements of signal, thereby overcoming the lack in spatial resolution⁸.

8. Line 110. Effective pump flow rate was not measurable.

In future work, consider using Transonic flow probes. They have nice in line flow probes or wrap around (vessel or tubing).

The authors thank the reviewer for this advice and are considering these flow probes for even more complete models in future research.

9. Was flow steady or pulsatile? If pulsatile, close to human heart rate? Pointing this out in terms of if pulsatile flow impacts DSA (likely minimal) and/or MPI.

Although a pulsatile pump was available in our lab, we deliberately chose a pump with steady flow as operation is easier and we did not consider pulsatile flow necessary to reach the goals of the experiment. However, we consider tracking pulsatile flow a main topic of future studies.

10. Line 128, is the 5 mm resolution isotropic?

Yes, this spatial resolution is isotropic in the 2D plane. The field-free line approach delivers a projection two-dimensional image as in DSA.

11. Line 129, need a period after used and ref 33 and before For...

This has been corrected.

DISCUSSION

12. Overall: Good that basic SNR measurements were provided. But what about geometric measurements? those are the most important, along with functional measurements like flow rate (e.g., typically by Doppler), when deciding on when to intervene for PAD as well as whether the interventional procedure has successfully re-opened the vessel. Do you currently have the resolution for any of this?

The authors consider the points raised by the reviewer very important. Detection of significant stenosis based on visual assessment could not be tested, because the cadavers did not have any. Temporal resolution on the other hand is plenty to detect flow-limitation by a significant stenosis, even if it cannot be differentiated visually.

This is often the case for DSA as well, when calcifications or post-procedural dissections obstruct the contrasted lumen and decisions must be made on contrast dynamics alone. Furthermore, sensitivity might be high enough to detect stenosis based on reduced MPI signal, which could be demonstrated in previous work by Herz et al (2018). The discussion section has been updated accordingly.

13. Related note: Ah, the main point of this work is what is done once the decision to intervene has been made. So, how about adding in the Intro a brief description of the

clinical work up prior to DSA and revascularization, e.g. often Ultrasound (geometric and doppler). This would help further define where MPI fits in.

The authors appreciate the notions of the reviewer to provide more background and have amended the introduction accordingly. However, we would like to add, that application of MPI in its current form does not need to be limited to interventions but could be used for diagnostics as well.

14. Lines 157-9: This study demonstrates the usability of a human-sized MPI scanner for vascular interventions under realistic conditions, which produces images resembling DSA to assess femoral vessels.

Please consider: "This study provides a proof of concept ~~usability~~ study of a human-sized ...". okay Usability studies are so important for eventual clinical adoption. Too many times scientists and engineers develop stuff that clinicians or the workflow itself just cannot use. But a true and thorough usability study does go beyond what is presented here.

We totally agree and have toned down this notion accordingly.

Regarding 'images resembling DSA' ... a little of an overstatement? Starting with Figure 4 ... are the MPI images the same length as displayed in the concomitant DSA? If not, please put a red dotted line (or some other demarcation around the DSA that the MPI is representing. If so, resembling is a bit of an over statement. Could you provide an order of magnitude for 'resembling' for important metrics? e.g. FOV, scan time, recon time, resolution. This could be rough quantifiable estimate, or even a qualitative assessment. But some sort of qualifier seems more appropriate. It is appreciated that the authors started with the term 'resembling' (vs. something more bombastic). In terms of temporal resolution and visual contrast

We totally agree and have added the qualifiers image contrast and temporal resolution to further specify, what was meant.

15. Lines 178-183, labeled is mis-spelled a couple times.

We were using the British spelling but have changed to US spelling.

16. Line 184: “diagnostic images”

Similar to comment above about images resembling DSA. This appears to be a vast overstatement, at this time. The current figures qualitatively do not look like they would be diagnostic needs, and there are no measurements such as geometry or flow. One way to satisfy this in the future could be to get feedback from the 'customer', i.e. interventional radiologist. For now, please soften this claim.

The authors fully agree and have toned down substantially.

17. Lines 187-88: Please comment on what causes the difference in SNR for Resotran vs Perimag? Solely based on size? Does one IO clump? Clear faster? What about the IO characteristics and/or interaction with mag fields of MPI result in not insignificant differences in SNR?

Thank you for this question. You are raising an important point there. We would like to take the chance and further elucidate on the matter. Resotran was designed with MRI in mind. It has a core diameter of around 7-9 nm, optimized for maximum T2 shortening effect. Apparently, Resotran works for MPI because some clustering occurs, which realizes an effective core diameter in the range of approx. 25 nm. This results in a usable signal for MPI. Perimag on the other hand was designed and

optimized for MPI with a core size of about 30 nm. The differences in SNR between Resotran and Perimag are in fact perfectly in line with the results of other studies^{9,10}.

18. Line 190, extra comma after suggests and before that.

Done.

19. Lines 192-206: So would it be up to 6 vials of Resotran, if one vial = single bolus? please clarify.

This is correct. Based on the current scanner performance, the assumptions stated in the manuscript and according to the results of the Phase II trial, using 6 vials of Resotran (1.5 ml each) could be safe, which would result in the possibility to acquire imaging series from 6 bolus injections.

Also comment on potential adverse events due to iron overload if this, or another clinically relevant administration regime, were to be used.

Ferucarbotran is generally well tolerated with a substantial body of evidence from clinical use in the context of MRI. However, several side effects have been reported. Common reactions include injection site reactions, which are not usually attributed to Ferucarbotran itself. Furthermore, transient changes in vital signs including mildly increased blood pressure and heart rate, which typically resolve minutes after injection, can occur. There are only anecdotal reports about allergic reactions, which have been described as mild. A small percentage of patients reported back pain during intravenous infusion for MRI, which can be largely avoided, when the solution is warmed to room temperature (own experiences and vendor statements). Clinically insignificant changes in liver enzymes or clotting time occasionally occur. One vial of Resotran contains 42 mg of iron, which is less than the maximum suggested daily intake according to the FDA (45 mg). Overall, the side effect profile is very favorable.

Nevertheless, future research will aim to increase sensitivity, so that doses above the approved levels will not be necessary.

20. Line 206: You mention sensitivity ... but what about resolution? From the figures comparing DSA to MPI, it appears resolution would be an equal or greater limiting factor right now. And is it easier/harder to overcome resolution challenges, if they are ~equal to sensitivity challenges?

Resolution and sensitivity are closely connected in MPI. Indeed, at the current stage of development, limitations in spatial resolution are harder to overcome. Spatial Resolution is determined by gradient strength, scan frequency and FFL velocity, and the particle system, among others. In short, the spatial resolution $FHWM$ is directly proportional to the temperature T and antiproportional to the gradient strength G and the particle parameters diameter D^3 and magnetization M_s ($FHWM \propto T/(G \cdot D^3 \cdot M_s)$).

This means, increasing the particles and/or the gradient strength enhances the spatial resolution. But on the other side, the speed of the encoding scheme, here the FFL, is important, which results in a higher SNR with increased frequency but on the other side relaxation effects of the particles can cause blurring effects at high frequencies. In addition, higher gradients requires higher magnetic fields. In combination with high frequencies, the PNS and/or SAR limitations have to be kept be kept in mind, which are the major limiting factors in the design of human sized MPI scanners. Thus, hardware limitations are incurred by the large bore size, PNS, SAR limitations, and limited selection of available tracers. More on this can be found in previous work ^{11,12}.

However, under the given circumstances, improved tracer systems seem to have the highest potential to increase sensitivity and spatial resolution to sub-millimeter levels

but this is ongoing research. (refer comments to Reviewer 2 and Tay, Z. W. *et al.* Superferromagnetic Nanoparticles Enable Order-of-Magnitude Resolution & Sensitivity Gain in Magnetic Particle Imaging. *Small Methods* **5**, e2100796, doi:10.1002/smt.202100796 (2021))

21. Lines 209-210: “An exact determination of the discriminatory resolution was hence not available.”

Why not? Couldn't you use the same DSA/MPI set up and use resolution phantoms? Dorenzo not ideal, something more like an artery so something cylindrical with different diameters. An extra experiment is not necessary, per se, because the images in the figures do give the reader a clear sense of differences in spatial resolution between DSA and MPI, but this sort of resolution experimental set up is important and likely to be used multiple times throughout the further development of this MPI application.

Thank you for raising this very important point. For the cadaver (we assume, this was meant by “Dorenzo”?) no ground truth was readily available. We fully agree, that a resolution phantom for validation studies would be expedient and this is currently under development. However, in this manuscript and context, the emphasis was put on practical aspects and the authors feel, that the topic of resolution warrants a dedicated experiment and manuscript.

22. Line 210: spelling error, should be discriminatory

Corrected.

23. Regarding higher temporal resolution of MPI: This is really a great benefit. Can

you please briefly add some commentary on how it is possible the MPI is faster than the DSA? Even with an extensive background in engineering and imaging science, across multiple modalities and preclinical and clinical, I do not intuitively know what the primary factors are here. Is it time to acquire x-rays, for example?

Reconstruction? Other?

This is a very important point that you raise here. DSA image acquisition is technically limited by detector readout and tube output. However, these limitations are hardly reached because radiation protection constraints prohibit utilizing their full potential (ALARA principle) and readout constraints are not disclosed by the manufacturer. Furthermore, higher frame rates are actually not necessary for practical purposes. On the other side, MPI can be a lot faster than what is currently utilized ¹³. As mentioned in comment 20, the temporal resolution in MPI scanners is defined by the frequencies the encoding scheme, FFP or FFL, is moved through the desired FOV. E.g., to cover an entire image within the iMPI scanner, the lowest frequency, here 60Hz, defines the coverage of a single frame. Thus, in $16.7 \text{ ms} / 2 = 8.4 \text{ ms}$ (divided by 2 because of two encoding cycles) a single image is acquired. Increasing the 'lowest' frequency, e.g, in a case for a small animal scanner with frequencies 1kHz and 14.25kHz, within $500 \mu\text{s}$ a full image can be acquired. However, as mentioned above, especially PNS regularities limits the frequencies but also the need for clinical purposes (~4 frames per second).

24. Line 256. Please add acoustic in front of noise here, to explicitly differentiate from signal/image noise.

Done.

25. Line 286: spelling error, should be spatial

Done.

26. Lines 287-88: But is it cost effective to layer on MPI? If so, to get rid of how much radiation dose? If you happen to have any information about these considerations, please consider adding here. But definitely needed for future immediate next steps in this argument and work.

We would like to answer this together with the below question.

27. Lines 292-3: Related to above, have you done rough calculations on % reduction of radiation possible, with the use of MPI instead (either full replacement, or as you also highlight partial replacement of DSA procedures)? Approximately... up to 90% of radiation dose per endovasc procedure are related to dsa, which could be ... by mpi. However, the aim of the current development is to augment dsa and not replace.

You are indeed raising another very good point here. Unfortunately, we have no data readily available on the exact amount that DSA or fluoroscopy contribute to total dose. However, based on personal experience, the ratio is roughly 80:20 for DSA. Furthermore, again speaking from personal experience, approximately 70% of all X-ray usage in an intervention require no high spatial or temporal resolution (i.e. survey angiography of otherwise unremarkable segments and advancing guidewires or stents to the target). As a rough estimate and in consideration of national reference values from Germany, which rate peripheral interventions at 10 mSv, we assume, that MPI could substitute on average up to 7 mSv per procedure. In Europe in

general and in Germany in particular, there is a strong emphasis on radiation protection. Nevertheless, we are fully aware that this is different in other regions of the world (i.e. US or Asia), which would completely change cost-benefit calculations. We tried to incorporate parts of this huge topic into the manuscript and hope, that this might partly answer the raised question. We fully agree, that this topic needs further in-depth coverage in future publications.

28. Line 298: “tracers in development”. Again, just a note, have to be careful here. ROA is critical. IV administration has way more risks than subq. It is good that there are, once again, approved IOs that have ROA of venous. Going straight arterial will likely require additional testing.

The authors fully agree with this very important remark and this is, what Reviewer #2 also argued. In consideration of this, please refer to our reply there and note, that we have augmented the limitations section substantially in this regard. Further research is certainly warranted to confirm the safe application of Resotran via intraarterial injection.

29. Line 358: Than should be then

Thank you for highlighting this spelling mistake, which we have corrected in the manuscript.

30. Line 415: mean maximum SI

Please refer answer to comment 31.

31. What is mean maximum? Were there multiple maximum SI measurements made and they were averaged?

Thank you for hinting at this rather ambiguous phrasing. To obtain the temporal maximum SI, we averaged the maximum SI over time from the proximal and distal ROIs (see Figure 5). We have rephrased the Methods section accordingly.

32. Line 615: Spelling error should be dotted.

Unfortunately, we were not able to determine a spelling error in this line, please clarify.

However, the whole manuscript was meticulously searched for spelling errors before final submission of the revised manuscript and several other spelling errors have been corrected.

References:

1. Mohn F, Scheffler K, Ackers J, et al. Characterization of the clinically approved MRI tracer resotran for magnetic particle imaging in a comparison study. *Phys Med Biol*. 2024;69(13). doi:10.1088/1361-6560/ad5828
2. Hartung V, Günther J, Augustin AM, et al. Resotran® meets MPI – clinically approved Ferucarbotran reintroduced: a major leap towards MPI in humans. *Int J Magn Part Imaging IJMPI*. 2023;9(1 Suppl 1). doi:10.18416/IJMPI.2023.2303058
3. Vogel P, Markert J, Rückert MA, et al. Magnetic Particle Imaging meets Computed Tomography: first simultaneous imaging. *Sci Rep*. 2019;9(1):12627. doi:10.1038/s41598-019-48960-1
4. Reimer P, Rummeny EJ, Daldrup HE, et al. Clinical results with Resovist: a phase 2 clinical trial. *Radiology*. 1995;195(2):489-496. doi:10.1148/radiology.195.2.7724772
5. Song G, Kenney M, Chen YS, et al. Carbon-coated FeCo nanoparticles as sensitive magnetic-particle-imaging tracers with photothermal and magnetothermal properties. *Nat Biomed Eng*. 2020;4(3):325-334. doi:10.1038/s41551-019-0506-0
6. Cutler JI, Zheng D, Xu X, Giljohann DA, Mirkin CA. Polyvalent Oligonucleotide Iron Oxide Nanoparticle “Click” Conjugates. *Nano Lett*. 2010;10(4):1477-1480. doi:10.1021/nl100477m
7. Tay ZW, Savliwala S, Hensley DW, et al. Superferromagnetic Nanoparticles Enable Order-of-Magnitude Resolution & Sensitivity Gain in Magnetic Particle Imaging. *Small Methods*. 2021;5(11):e2100796. doi:10.1002/smt.202100796

8. Herz S, Vogel P, Kampf T, et al. Magnetic Particle Imaging for Quantification of Vascular Stenoses: A Phantom Study. *IEEE Trans Med Imaging*. 2018;37(1):61-67. doi:10.1109/TMI.2017.2717958
9. Eberbeck D, Dennis CL, Huls NF, Krycka KL, Gruttner C, Westphal F. Multicore Magnetic Nanoparticles for Magnetic Particle Imaging. *IEEE Trans Magn*. 2013;49(1):269-274. doi:10.1109/TMAG.2012.2226438
10. Vogel P, Kampf T, Rückert M, et al. Synomag®: The new high-performance tracer for magnetic particle imaging. *Int J Magn Part Imaging IJMPI*. 2021;7(1). doi:10.18416/IJMPI.2021.2103003
11. Vogel P, Ruckert MA, Greiner C, et al. iMPI: portable human-sized magnetic particle imaging scanner for real-time endovascular interventions. *Sci Rep*. 2023;13(1):10472. doi:10.1038/s41598-023-37351-2
12. Saritas EU, Goodwill PW, Croft LR, et al. Magnetic particle imaging (MPI) for NMR and MRI researchers. *J Magn Reson San Diego Calif 1997*. 2013;229:116-126. doi:10.1016/j.jmr.2012.11.029
13. Vogel P, Rückert MA, Kampf T, et al. Superspeed Bolus Visualization for Vascular Magnetic Particle Imaging. *IEEE Trans Med Imaging*. 2020;39(6):2133-2139. doi:10.1109/TMI.2020.2965724

Andreia Cunha, PhD
Chief Editor
Communications Medicine

Dear Ms Cunha, dear editors of Communications Medicine,

We would like to thank you for the initial feedback and hereby resubmit our revised manuscript "**Magnetic particle imaging angiography of the femoral artery in a human cadaveric perfusion model**" (COMMSMED-24-0667B).

We would like to thank you again for the time and effort invested in helping improve our work, which will certainly contribute to promoting the field of Magnetic Particle Imaging. The three reviewers' comments have been thoroughly addressed in the previous point-by-point response and were acknowledged by the comment of Reviewer #2 concerning the amended version.

We hope that we were able to improve our work sufficiently to merit publication in Communications medicine.

Sincerely yours,

Viktor Hartung

COMMSMED-24-0667B

Magnetic particle imaging angiography of the femoral artery in a human cadaveric perfusion model

Response to Reviewers

The authors would like to thank all of the reviewers for their valuable feedback on our manuscript and greatly appreciate helping to improve our work. We thoroughly addressed all comments in the final manuscript.

Reviewer #2

The revised manuscript has addressed our concerns and we have no further comments for author.

Your acknowledgement of our work is much appreciated. Thank you again for the initial feedback which allowed us to improve our work.